# Environmental and Pathogenic Carbapenem Resistant Bacteria Isolated from a Wastewater Treatment Plant Harbour Distinct Antibiotic Resistance Mechanisms

**DOI:** 10.3390/antibiotics10091118

**Published:** 2021-09-16

**Authors:** Micaela Oliveira, Inês Carvalho Leonardo, Mónica Nunes, Ana Filipa Silva, Maria Teresa Barreto Crespo

**Affiliations:** 1Instituto de Tecnologia Química e Biológica António Xavier, Universidade Nova de Lisboa, Avenida da República, 2780-157 Oeiras, Portugal; 2iBET, Instituto de Biologia Experimental e Tecnológica, Apartado 12, 2781-901 Oeiras, Portugal; micaelaoliveira@ibet.pt (M.O.); ines.leonardo@ibet.pt (I.C.L.); tcrespo@ibet.pt (M.T.B.C.); 3Section of Microbiology, Department of Biology, University of Copenhagen, Universitetsparken 15, DK-2100 Copenhagen, Denmark; anafcsilva@gmail.com

**Keywords:** antibiotic resistance, carbapenems, wastewater treatment plants, discharged effluents, environmental and pathogenic carbapenem resistant bacteria

## Abstract

Wastewater treatment plants are important reservoirs and sources for the dissemination of antibiotic resistance into the environment. Here, two different groups of carbapenem resistant bacteria—the potentially environmental and the potentially pathogenic—were isolated from both the wastewater influent and discharged effluent of a full-scale wastewater treatment plant and characterized by whole genome sequencing and antibiotic susceptibility testing. Among the potentially environmental isolates, there was no detection of any acquired antibiotic resistance genes, which supports the idea that their resistance mechanisms are mainly intrinsic. On the contrary, the potentially pathogenic isolates presented a broad diversity of acquired antibiotic resistance genes towards different antibiotic classes, especially β-lactams, aminoglycosides, and fluoroquinolones. All these bacteria showed multiple β-lactamase-encoding genes, some with carbapenemase activity, such as the *bla*_KPC_-type genes found in the *Enterobacteriaceae* isolates. The antibiotic susceptibility testing assays performed on these isolates also revealed that all had a multi-resistance phenotype, which indicates that the acquired resistance is their major antibiotic resistance mechanism. In conclusion, the two bacterial groups have distinct resistance mechanisms, which suggest that the antibiotic resistance in the environment can be a more complex problematic than that generally assumed.

## 1. Introduction

The increasing dissemination of carbapenem resistant bacteria represents a major worldwide problem and an important threat to human health [1]. Despite this being a concern generally associated with health care facilities, the presence of carbapenem resistant bacteria is also increasing in wastewater treatment plants (WWTPs), which are already considered relevant anthropogenic reservoirs and sources for the spread of the antibiotic (AB) resistance into the environment [2,3,4].

Carbapenem resistance can result from both enzyme-mediated and/or non-enzyme-mediated processes of bacteria [1,5]. The enzyme-mediated resistance mechanisms are encoded by specific *bla* genes and involve the hydrolysis of these ABs by carbapenemases, a particular group of β-lactamases that hydrolyse not only carbapenems but also other important β-lactam ABs, such as penicillins, cephalosporins, and monobactams [1,5,6,7]. In terms of clinical relevance and global distribution, the most important carbapenemases are: (1) Class A serine β-lactamases, encoded by *bla*_KPC_-type genes; (2) Class B metallo-β-lactamases, encoded by *bla*_NDM_, *bla*_IMP_, and *bla*_VIM_-type genes; and (3) Class D serine-β-lactamases, encoded by *bla*_OXA-48_-type genes [5,8]. These carbapenemase-encoding genes can be found in the bacterial chromosome, but more often in conjugative plasmids, which promotes their horizontal transfer between resistant and non-resistant bacteria. In fact, the plasmid transfer and acquisition is the main driver of the rapid increase and global spread of the carbapenem resistance that has been observed in the last decade [1,5,8,9]. However, bacteria can also be resistant to carbapenems due to mutations causing loss of expression of porin-encoding genes; as a result of the overexpression of genes encoding for efflux pumps; or due to mutations that modify the production levels or the binding affinities of the penicillin-binding proteins [1,5]. These non-enzyme mediated resistance mechanisms, also known as intrinsic resistance mechanisms, can occur alone or together with the production of extended-spectrum β-lactamases, cephalosporinases, and/or carbapenemases, generating well-known carbapenem resistance phenotypes [1,5].

Several recent studies already point out the existence of high concentrations of carbapenemase-encoding genes and corresponding carbapenem resistant bacteria along all the main steps of different wastewater treatment processes worldwide, from the wastewater influents to the treated effluents, warning of their subsequent release into the water bodies [10,11,12,13,14,15]. Nevertheless, most of these studies have especially focused on the detection and quantification of target genes by PCR/qPCR techniques. This has led to a serious gap in the isolation of the different populations of carbapenem resistant bacteria that exist in the wastewater environments and on the assessment and characterization of their acquired resistance genes and intrinsic resistance mechanisms towards carbapenems and other ABs. The work developed by Hrenovic et al., [16,17,18,19] explores the isolation of different populations of carbapenem resistant bacteria from the wastewater environments using two incubation temperatures in selective culture media: (1) Incubation at 37 °C for the isolation of presumably environmental carbapenem resistant bacteria, whose resistance mechanisms are thought to be mainly intrinsic; (2) Incubation at 42 °C for the isolation of presumably pathogenic carbapenem resistant bacteria, whose resistance mechanisms are thought to be mainly acquired. However, these studies mostly address the physicochemical characterization of the different wastewater environments and the abundance of these populations of carbapenem resistant bacteria along the wastewater treatment processes. Therefore, a deeper genotypical and phenotypical characterization of environmental and pathogenic isolates is still crucial for a better understanding of these increasingly important AB resistance reservoirs.

Accordingly, the aim of the present study was to genotypically and phenotypically characterize the AB resistance profile of several environmental and pathogenic carbapenem resistant bacteria isolated from a full-scale WWTP.

## 2. Results

### 2.1. Concentrations of Carbapenem Resistant Bacteria

Potentially environmental and potentially pathogenic carbapenem resistant bacteria were detected at high concentrations in wastewater influent and discharged effluent samples (Appendix A). The potentially environmental carbapenem resistant bacteria—isolated after incubation at 30 °C—presented concentrations of 9.58 × 10^4^ CFU/mL in the wastewater influent and 5.37 × 10^3^ CFU/mL in the discharged effluent, whereas the potentially pathogenic carbapenem resistant bacteria—isolated after incubation at 42 °C—presented concentrations of 1.38 × 10^3^ CFU/mL in the wastewater influent and 1.20 × 10^2^ CFU/mL in the discharged effluent (Appendix A). All bacterial concentrations correspond to the mean of the mean values obtained for the technical triplicates of each of the three biological samples collected for each sampling point.

### 2.2. Species Identification by 16S rRNA Gene Sequencing and Screening of Carbapenem Resistance Genes

Thirty-two carbapenem resistant isolates were obtained from wastewater influent samples incubated at 30 °C. The taxa identified were: *Pseudomonas* spp. (*n* = 16), namely the species *P. entomophila* (1), *P. fluorescens* (3), *P. fragi* (1), *P. lundensis* (1), *P. migulae* (1), *P. psychrophila* (1), *P. putida* (7), and *P. syringae* (1); *Aeromonas* spp. (*n* = 15), namely the species *A. caviae* (7), *A. salmonicida* (1), and *A. veronii* (7); and *Raoultella ornithinolytica* (*n* = 1) (Appendix A). The screening of carbapenem resistance genes revealed the presence of *bla*_KPC_-type genes in eight of these bacterial isolates (Appendix A). Twenty-six carbapenem resistant isolates were obtained from the discharged effluent samples incubated at 30 °C. The taxa identified were: *Pseudomonas* spp. (*n* = 13), namely the species *P. entomophila* (1), *P. fluorescens* (2), *P. fragi* (2), *P. monteilii* (1), *P. psychrophila* (1), and *P. putida* (6); *Aeromonas* spp. (*n* = 9), namely the species *A. caviae* (1) and *A. veronii* (8); *Chromobacterium rhizoryzae* (*n* = 3); and *Acinetobacter pittii* (*n* = 1) (Appendix A). In the screening of carbapenem resistance genes, there was no detection of any of the five target resistance genes in those bacterial isolates (Appendix A).

For the incubation at 42 °C, twenty-seven carbapenem resistant isolates were obtained from the wastewater influent samples. The taxa identified were: *Acinetobacter* spp. (*n* = 11), namely the species *A. baumannii* (6) and *A. pittii* (5); *Escherichia coli* (*n* = 6); *Citrobacter* spp. (*n* = 3), namely the species *C. amalonaticus* (1) and *C. freundii* (2); *Klebsiella* spp. (*n* = 3), namely the species *K. pasteurii* (1) and *K. pneumoniae* (2); *Enterobacter asburiae* (*n* = 2); and *Raoultella ornithinolytica* (*n* = 2) (Appendix A). The screening of carbapenem resistance genes revealed the presence of *bla*_KPC_-type genes in nineteen bacterial isolates, *bla*_OXA-48_-type genes in two bacterial isolates, and *bla*_VIM_-type genes in seven bacterial isolates (Appendix A). Twenty-two carbapenem resistant isolates were obtained from the discharged effluent samples. The taxa identified were: *Klebsiella pneumoniae* (*n* = 8); *Acinetobacter* spp. (*n* = 6), namely the species *A. baumannii* (5) and *A. pittii* (1); *Escherichia coli* (*n* = 4); *Aeromonas veronii* (*n* = 2); and *Citrobacter* spp. (*n* = 2), namely the species *C. amalonaticus* (1) and *C. freundii* (1) (Appendix A). The screening of carbapenem resistance genes revealed the presence of *bla*_KPC_-type genes in fourteen bacterial isolates and *bla*_VIM_-type genes in one bacterial isolate (Appendix A).

After both the species identification and the screening of carbapenem resistance genes in the bacterial isolates obtained from the discharged effluent samples, ten of the bacterial isolates grown at 30 °C and seven of bacterial isolates grown at 42 °C (one bacterial isolate from each of the species identified for both incubation temperatures) were selected to perform a whole genome sequencing analysis.

### 2.3. Taxonomic Confirmation Using the Whole Genome Sequencing Data

Both the SpeciesFinder 2.0 and the KmerFinder 3.2 tools positively confirmed the genus of the ten bacterial isolates grown at 30 °C (Appendix A). Regarding the species level, the SpeciesFinder 2.0 tool confirmed the identification of three of these isolates and the KmerFinder 3.2 tool confirmed the identification of seven of these isolates (Appendix A). For the identification of the bacterial isolates grown at 42 °C, the SpeciesFinder 2.0 tool positively confirmed the genus of six bacterial isolates and the species of four of these isolates, whereas the KmerFinder 3.2 tool positively confirmed the genus of the seven bacterial isolates and the species of five of these isolates (Appendix A). The taxonomic identifications obtained with the KmerFinder 3.2 tool will be used from now on in this article, since this is considered to be the most accurate tool for the identification of different bacterial strains [20]. Therefore, the *Pseudomonas psychrophila*, *Pseudomonas putida,* and *Acinetobacter pittii* isolates—grown at 30 °C—will now be renamed as *Pseudomonas fragi*, *Pseudomonas* sp. URMO17WK12:I11 and *Acinetobacter oleivorans*, respectively, and the *Citrobacter amalonaticus* and *Citrobacter freundii* isolates—grown at 42 °C—will now be renamed as *Citrobacter* sp. Y3 and *Citrobacter portucalensis*, respectively.

### 2.4. Identification of Acquired AB Resistance Genes, Conjugative Plasmids, and Virulence Factors

#### 2.4.1. Acquired AB Resistance Genes

After the analysis using both the ResFinder 4.0 and KmerResistance 2.2 tools, three bacterial isolates grown at 30 °C, namely the *A. oleivorans*, *A. caviae,* and *A. veronii* isolates, presented different acquired AB resistance genes, including acquired β-lactam resistance genes, in their whole genomes (Figure 1; Appendix A). Besides the presence of β-lactam resistance genes, aminoglycoside, fluoroquinolone, macrolide, phenicol, rifampicin, sulphonamide, and trimethoprim resistance genes were also detected. However, in the remaining seven bacterial isolates grown at 30 °C—*C. rhizoryzae*, *P. entomophila*, *P. fluorescens*, *P. fragi* (2), *P. monteilii,* and *Pseudomonas* sp. URMO17WK12:I11—there was no detection of any acquired AB resistance genes (Figure 1; Appendix A).

Regarding the bacterial isolates grown at 42 °C, the analysis performed with the ResFinder 4.0 and KmerResistance 2.2 tools revealed that all bacterial isolates, namely the *A. baumannii*, *A. pittii*, *A. veronii*, *C. portucalensis*, *Citrobacter* sp. Y3, *E. coli,* and *K. pneumoniae* isolates, presented different acquired AB resistance genes, including acquired β- lactam resistance genes, in their whole genomes (Figure 1; Appendix A). Besides the presence of the β-lactam resistance genes, aminoglycoside, colistin, fluoroquinolone, fosfomycin, macrolide, phenicol, sulphonamide, tetracycline, and trimethoprim resistance genes were also abundantly detected. In fact, a total of about 71% of the bacterial isolates grown at 42 °C presented acquired resistance genes to, at least, three different AB classes (Figure 1; Appendix A).

#### 2.4.2. Conjugative Plasmids in *Enterobacteriaceae* spp. and Virulence Factors in *E. coli*

According to the PlasmidFinder 2.0 tool, which identifies plasmids in total or partial sequenced isolates of *Enterobacteriaceae* spp., all carbapenem resistant *Enterobacteriaceae* isolated from the discharged effluent samples—*C. portucalensis*, *Citrobacter* sp. Y3, *E. coli* and *K. pneumoniae*—harbour different plasmids known to contain the previously described acquired AB resistance genes (Figure 2). Moreover, the VirulenceFinder 2.0 tool, which identifies virulence genes in sequenced *E. coli* isolates, revealed the presence of the *ast*A, *gad*, *iss*, *lpf*A, *omp*T, *sit*A, *ter*C, and *tra*T virulence factors in the whole genome of the *E. coli* isolate (Figure 2).

### 2.5. AB Susceptibility Testing

Apart from the *A. caviae* and *A. veronii* (2) isolates, for which there are no EUCAST breakpoints available for most of the ABs used, all bacterial isolates, namely *A. baumannii*, *A. oleivorans*, *A. pittii*, *C. portucalensis*, *Citrobacter* sp. Y3, *E. coli,* and *K. pneumoniae*, showed resistance phenotypes towards ampicillin and cefotaxime and, apart from the *A. oleivorans* and *A. pittii*, to imipenem and meropenem (Table 1). Besides the resistance phenotypes to the β-lactam class, all bacterial isolates presented resistance phenotypes to fluoroquinolones (ciprofloxacin) and tetracyclines (tetracycline) (Table 1). Moreover, most bacterial isolates also showed resistance phenotypes towards chloramphenicol, gentamicin, trimethoprim, and trimethoprim + sulphamethoxazole (Table 1). All bacterial isolates showed resistance phenotypes to more than three AB classes, being considered multi-resistant.

## 3. Discussion

The presence of high concentrations of carbapenem resistant bacteria in both the wastewater influent and discharged effluent samples is not only a reflection of the increasing use of carbapenems in recent years, but also an additional evidence regarding the inefficiency of the conventionally applied wastewater treatments in the elimination of these microorganisms from the treated effluents, subsequently leading to their release into the environment [13].

The incubation of the wastewater influent and discharged effluent samples in selective media at 30 °C and 42 °C allowed the isolation and characterization of two different populations of carbapenem resistant bacteria: the potentially environmental and the potentially pathogenic [16,17,18,19]. At 30 °C, it is expected that both grow. However, and given the environmental nature of the samples, the proportion of environmental bacteria will be much higher than the proportion of pathogenic bacteria. Thus, when randomly picking colonies, the odds of picking an environmental carbapenem resistant bacteria are much higher than the odds of picking a pathogenic carbapenem resistant bacteria. Concordantly, and despite the possible bias induced by this methodology, in which pathogenic carbapenem resistant bacteria are also able to grow in the environmental carbapenem resistant bacteria plates, at 30 °C, most of the identified bacteria were described as having an environmental origin, although three potentially pathogenic bacteria for both humans and animals were also isolated. In general, these environmental carbapenem resistant bacteria presented very low detection rates of *bla*_KPC_, *bla*_OXA-48_, *bla*_NDM_, *bla*_IMP_, and *bla*_VIM_-type genes, raising the hypothesis that most of them could be intrinsically resistant to these ABs. On the contrary, most of the identified bacteria at 42 °C were well-known human and animal pathogens. These bacteria presented much higher detection rates of the target carbapenem resistance genes, suggesting that they are resistant to carbapenems mainly as a result of the expression of these and other acquired resistance genes.

To have a deeper knowledge on the distinct resistance mechanisms of both the potentially environmental and potentially pathogenic carbapenem resistant bacteria obtained from the discharged effluent samples, the genotype of a bacterial isolate from each species obtained at 30 °C and 42 °C was characterized through a whole genome sequencing analysis, which, in a first approach, allowed the taxonomic confirmation of all bacteria. The whole genome sequencing data revealed that the potentially environmental carbapenem resistant bacteria did not harbour any acquired AB resistance genes, which corroborates the previous *TaqMan* multiplex qPCR results and supports the idea that their resistance to carbapenems may result from intrinsic mechanisms. This could be explained by the occurrence of mutations causing loss of expression of porin-encoding genes, the overexpression of genes encoding for efflux pumps, or the occurrence of mutations that modify the production levels or the binding affinities of the penicillin-binding proteins [1,5]. In fact, studies are starting to report that the majority of the carbapenem resistant bacteria of environmental origin present in aquatic environments, namely *Chromobacterium* spp. and *Pseudomonas* spp. isolates, have different intrinsic mechanisms of AB resistance [12,16,17,18,19,21]. Generally, these bacteria are not considered of epidemiological relevance since most of them are very unlikely to generate human or animal infections and to horizontally transfer AB resistance genes to other bacteria [12]. For the potentially pathogenic carbapenem resistant bacteria isolated from the discharged effluent samples (three isolated at 30 °C—it happened despite the odds favouring the pick of environmental carbapenem resistant bacteria—and seven isolated at 42 °C), the whole genome sequencing data revealed the presence of a vast diversity of acquired AB resistance genes. Most of these bacteria represent important human and animal pathogens, able to cause a variety of nosocomial infections with frequent negative patient outcomes. In fact, the World Health Organization has already classified the carbapenem resistant *A. baumannii*, *E. coli,* and *K. pneumoniae* as microorganisms of critical priority for the research, discovery, and development of new ABs due to their increasing spread worldwide and to the urgent need of new effective treatments [22]. These ten bacterial isolates presented multiple acquired β-lactamase-encoding genes, some of which with carbapenemase activity. Among the acquired carbapenemase-encoding genes, *bla*_KPC_-type genes were found in the four *Enterobacteriaceae* isolates (*C. portucalensis*, *Citrobacter* sp. Y3, *E. coli* and *K. pneumoniae*), which agrees with previous studies reporting the predominance of these genes in different clinical isolates of carbapenemase-producing *Enterobacteriaceae* already responsible for serious outbreaks in Portugal [23,24,25,26]. Concordantly, all four *Enterobacteriaceae* isolates also carry IncF or IncP-like plasmids, well known for their major roles in the dissemination not only of *bla*_KPC_-type genes, but also of other AB resistance genes among *Enterobacteriaceae* [24,27]. Besides the β-lactam resistance genes, acquired resistance genes towards other important AB classes, such as aminoglycosides and fluoroquinolones, were also present with high detection rates in the potentially pathogenic carbapenem resistant isolates, which was expected due to the common co-occurrence of β-lactam, aminoglycoside and fluoroquinolone resistance genes in the same conjugative plasmids [9,28]. Moreover, among the acquired AB resistance genes towards other AB classes found in some bacterial isolates, the most concerning situation was the detection of the newly identified *mcr*-9 gene variant, conferring resistance towards colistin and frequently present in IncHI2-like plasmids, in the *E. coli* isolate [29]. Since colistin is currently one of the most effective last-resort ABs used in the treatment of severe infections caused by carbapenem resistant bacteria, the rapid development and spread of the mobilized colistin resistance (*mcr*) genes represent another serious public health challenge [29,30]. In fact, *E. coli* isolate presented not only a great diversity of acquired AB resistance genes and conjugative plasmids, but also ten different virulence factors, having one of the most alarming genetic profiles among the bacterial isolates obtained from the discharged effluent samples. Altogether, the obtained results suggest that the great majority of the potentially pathogenic carbapenem resistant bacteria relies on the acquisition and expression of AB resistance genes as their main resistance mechanism, in opposition to the potentially environmental carbapenem resistant bacteria, which appear to be resistant to different ABs especially through intrinsic mechanisms, alone or in combination.

The analysis of the AB resistance phenotypes of the potentially pathogenic carbapenem resistant bacteria isolated from the discharged effluent samples revealed that all were multi-resistant, showing resistance phenotypes to more than three AB classes. These results reinforce the importance of the conjugative plasmids in the simultaneous dissemination of different AB resistance genes and is in line with the reported occurrence of multi-resistant bacteria in the Portuguese wastewater environments [31]. Furthermore, only the *A. oleivorans* and *A. pittii* bacterial isolates showed an intermediate susceptibility phenotype towards carbapenems (imipenem and meropenem), only the *A. oleivorans* bacterial isolate presented susceptibility to aminoglycosides (gentamicin) and only the *Citrobacter* sp. Y3 bacterial isolate was susceptible to trimethoprim. These results not only show that the acquired AB resistance genes are expressed and originate the corresponding resistance phenotypes, but also demonstrate that some of these bacteria also have intrinsic resistance mechanisms to ABs for which no acquired AB resistance genes were found in their whole genomes.

## 4. Materials and Methods

### 4.1. WWTP Description and Sample Collection

Samples were collected from a Portuguese full-scale WWTP designed to treat the domestic wastewater of approximately 756,000 population equivalents (P.E.) employing biological aerated filter technology. Two sampling points were defined, and three biological samples of 10 L each were collected from the wastewater influent and discharged effluent in sterile containers in November of 2019. After collection, all samples were directly transported to the laboratory under refrigerated conditions and immediately processed upon arrival.

### 4.2. Determination of Carbapenem Resistant Bacteria Concentrations

To determine the concentrations of carbapenem resistant bacteria in the wastewater influent and discharged effluent, ten-fold serial dilutions of each sampling point were made in 1 L sterile saline solution and 100 mL samples of each dilution were filtered in triplicate through sterile 0.22 µm pore-size polyethersulfone (PES) filters (Pall Corporation, New York, NY, USA). Then, the filters were placed on selective and chromogenic CHROMagar™ mSuperCARBA™ plates (CHROMagar, Paris, France) and incubated at 30 °C or 42 °C for 24 h. The different incubation temperatures were adapted from the work developed by Hrenovic et al., [16,17,18,19] and used to distinguish between potentially environmental bacteria, which are normally able to grow at 30 °C, and potentially pathogenic bacteria, which are usually also able to survive and grow at higher temperatures. Since many environmental bacteria have optimal growth temperatures below 37 °C, their isolation was performed at 30 °C instead of 37 °C, so that as few different species as possible would have their growth inhibited due to the temperature used. Following incubation, the colonies obtained on each plate were enumerated and their concentrations were calculated and expressed as colony-forming units per millilitre (CFU/mL).

### 4.3. Isolation, DNA Extraction, and Species Identification of Carbapenem Resistant Bacteria

A subset of colonies of each incubated plate showing different phenotypes were randomly picked and sub-cultured in the same medium and conditions until pure colonies were obtained. The pure colonies were inoculated in Tryptic Soy Broth (TSB) (VWR, Radnor, PA, USA) supplemented with 0.5 µg/mL meropenem trihydrate (Sigma-Aldrich, Saint Louis, MO, USA) and again incubated at 30 °C or 42 °C (depending on its original growth temperature) with shaking (150 rpm), overnight. After incubation, the bacterial DNA was extracted using the standard protocol from the DNeasy UltraClean Microbial Kit (Qiagen, Hilden, Germany) and the DNA concentrations and purities were measured using a NanoDrop 1000 Spectrophotometer (Thermo Fisher Scientific, Waltham, MA, USA). The extracted DNAs were then used for the species identification, which was performed by 16S rRNA gene sequencing. Briefly, the 16S rRNA genes from each colony were amplified by PCR with the primer pair Fw [5′-GCCAGCAGCCGCGGTAA-3′] and Rv [5′-AAGGAGGTGATCCRGCCGCA-3′] (adapted from [32]) in a mix reaction containing 12.5 µL NZYTaq II 2x Green MasterMix (NZYTech, Lisbon, Portugal), 200 nM of each forward and reverse primers, 50 ng of DNA template and nuclease free water (to complete 25 µL). The PCR reactions were conducted in a Doppio Termocycler (VWR, Radnor, PA, USA) using the following program: initial denaturation at 95 °C for 3 min; 35 cycles of amplification at 94 °C for 30 s, 57 °C for 30 s, and 72 °C for 45 s; final elongation at 72 °C for 5 min. The PCR products were then sent to Eurofins Genomics (Ebersberg, Germany) for purification and Sanger Sequencing and the resulting sequences were aligned against the National Centre for Biotechnology (NCBI) 16S rRNA gene database using the BLASTn algorithm to determine the taxonomic identities.

### 4.4. Screening of Carbapenem Resistance Genes

The extracted DNAs were also used to perform a screening of five important carbapenem resistance genes in terms of clinical relevance and global distribution—*bla*_KPC_, *bla*_OXA-48_, *bla*_NDM_, *bla*_IMP_ and *bla*_VIM_—by two previously developed *TaqMan* multiplex qPCRs [13]. The *TaqMan* multiplex qPCR 1 was designed for the detection and quantification of *bla*_KPC_ and *bla*_OXA-48_-type genes, whereas the *TaqMan* multiplex qPCR 2 was designed for the detection and quantification of *bla*_NDM_, *bla*_IMP,_ and *bla*_VIM_-type genes. The *TaqMan* multiplex qPCR assays were conducted in triplicate on a LightCycler 96 Real-Time PCR System (Roche, Basel, Switzerland) using the following program: DNA denaturation/polymerase activation at 95 °C for 5 min; 40 cycles of amplification at 95 °C for 10 s and 60 °C for 30 s. Information about the primers and probes is provided in Appendix A and the composition of the mix reactions is provided in Appendix A.

### 4.5. Whole Genome Sequencing and Assembly

Based on the species identification and on the screening of carbapenem resistance genes, the isolate with the biggest number and diversity of the target carbapenem resistance genes from each different species isolated from the discharged effluent samples at 30 °C and 42 °C was chosen to perform an enhanced whole genome sequencing service at MicrobesNG (Birmingham, UK), which combines two distinct technologies: the Illumina short reads and the Oxford Nanopore long reads.

For the Illumina sequencing, plated cultures of each bacterial isolate were inoculated into a cryopreservative (Microbank, Pro-Lab Diagnostics, Richmond Hill, ON, Canada). Then, between 10 and 20 microlitres of each suspension was lysed with 120 μL of Tris-EDTA (TE) buffer containing lysozyme (in a final concentration of 0.1 mg/mL) and RNase A (ITW Reagents, Barcelona, Spain) (in a final concentration of 0.1 mg/mL) and incubated at 37 °C for 25 min. After that, proteinase K (VWR, Radnor, USA) (in a final concentration of 0.1 mg/mL) and SDS (Sigma-Aldrich, Saint Louis, MO, USA) (in a final concentration of 0.5% *v*/*v*) were added and the mixture was incubated at 65 °C for 5 min. The genomic DNA was then purified using an equal volume of solid phase reversible immobilization beads, resuspended in EB buffer (Qiagen, Hilden, Germany) and quantified with the Quant-iT dsDNA HS Assay Kit (Thermo Fisher Scientific, Waltham, MA, USA) in an Eppendorf AF2200 plate reader (Eppendorf, Hamburg, Germany). The genomic DNA libraries were prepared on a Hamilton Microlab STAR automated liquid handling system (Hamilton Bonaduz AG, Rapperswil-Jona, Switzerland) using the Nextera XT DNA Library Prep Kit (Illumina, San Diego, CA, USA) according to the manufacturer’s instructions, with the following modifications: (1) 2 ng of DNA were used as input; (2) PCR elongation time was increased from 30 s to 1 min. The pooled libraries were quantified using the Kapa Biosystems Library Quantification Kit for Illumina (Roche, Basel, Switzerland) on a LightCycler 96 Real-Time PCR System (Roche, Basel, Switzerland) and then sequenced with the Illumina HiSeq (Illumina, San Diego, CA, USA) using a 250 bp paired end protocol.

For the Oxford Nanopore sequencing, broth cultures of each bacterial isolate were pelleted out, and then resuspended in a cryopreservative (Microbank, Pro-Lab Diagnostics, ON, Canada). Then, approximately 2 × 10^9^ cells were used for high molecular weight DNA extraction with the Nanobind CCB Big DNA Kit (Circulomics, Baltimore, MD, USA). The DNA was quantified with the Qubit dsDNA HS Assay (Thermo Fisher Scientific, Waltham, MA, USA) in a Qubit 3.0 Fluorometer (Thermo Fisher Scientific, Waltham, MA, USA) and the long read genomic DNA libraries were then prepared with both the Oxford Nanopore SQK-RBK004 Kit and/or with the SQK-LSK109 Kit with Native Barcoding EXP-NBD104/114 (Oxford Nanopore Technologies, Oxford, UK) using 400–500 ng of high molecular weight DNA. At the end, 12 to 24 barcoded samples were pooled together into a single sequencing library and loaded in a FLO-MIN106 (R.9.4 or R.9.4.1) flow cell in a GridION (Oxford Nanopore Technologies, Oxford, UK).

The Illumina reads were trimmed using Trimmomatic v. 0.30 [33] with a sliding window quality cut-off of Q15, the genome assemblies were performed with Unicycler v. 0.4.0 [34] and the contigs were annotated using Prokka v. 1.11 [35].

### 4.6. Taxonomic Confirmation and Identification of Acquired AB Resistance Genes, Conjugative Plasmids, and Virulence Factors

The taxonomic confirmation was conducted using the tools SpeciesFinder 2.0 and KmerFinder 3.2 [20]. Then, the identification of acquired AB resistance genes was performed using ResFinder 4.0 [36], with a threshold of 90% identity and a minimum length of 60%, and KmerResistance 2.2 [37,38], with thresholds of 70% identity and 10% depth corr. The detection of conjugative plasmids in the *Enterobacteriaceae* genomes was conducted using PlasmidFinder 2.0 [39] and the presence of virulence factors in the *Escherichia coli* isolates was assessed using VirulenceFinder 2.0 [40], with a threshold of 90% identity and a minimum length of 60%.

### 4.7. AB Susceptibility Testing

The AB resistance phenotypes of the potentially pathogenic carbapenem resistant bacteria isolated from the discharged effluent samples were determined according to the EUCAST disk diffusion method on Mueller–Hinton agar (BD Difco, Franklin Lakes, NJ, USA) [41]. From the ABs tested, four belong to the β-lactam class—ampicillin 10 µg (AMP10), cefotaxime 5 µg (CTX5), imipenem 10 µg (IMP10), and meropenem 10 µg (MEM10) (Oxoid, Ottawa, ON, Canada)—and six belong to AB classes for which different acquired resistance genes were found in the whole genomes of some bacterial isolates—chloramphenicol 30 µg (C30), ciprofloxacin 5 µg (CIP5), gentamicin 10 µg (CN10), tetracycline 30 µg (TE30), trimethoprim 5 µg (W5), and trimethoprim + sulphamethoxazole 25 µg (SXT25) (Oxoid, Ottawa, Canada). These ABs were chosen due to their wide use in the treatment of different bacterial infections. Triplicates of each bacterial isolate were adjusted to the 0.5 McFarland standard concentration and inoculated on Mueller–Hinton agar plates. Then, the antimicrobial disks were applied, and the plates were incubated at 35 ± 1 °C for 18 ± 2 h. After incubation, the inhibition zone diameters were measured and the bacterial isolates were categorized as “S” (susceptible, standard dosing regimen), “I” (susceptible, increased exposure), or “R” (resistant) [42]. Bacterial isolates presenting a resistance phenotype to, at least, three different AB classes were considered multi-resistant. The strain *Escherichia coli* ATCC 25922 was used as a control of these AB susceptibility testing assays.

## 5. Conclusions

This study allowed the isolation of two different populations of carbapenem resistant bacteria from the wastewater influent and discharged effluent of a full-scale WWTP—the potentially environmental and the potentially pathogenic—and the genotypical and phenotypical characterization of their AB resistance profile. It was possible to observe that: (1) Although with a reduction in the concentrations of CFU between the wastewater influent and discharged effluent, both potentially environmental and potentially pathogenic fractions of carbapenem resistant bacteria were present at high concentrations in the discharged effluent samples; (2) The potentially environmental carbapenem resistant bacteria presented low detection rates of acquired AB resistance genes, appearing to be resistant to carbapenems and to other ABs mainly through intrinsic mechanisms. (3) The potentially pathogenic carbapenem resistant bacteria presented high detection rates of acquired resistance genes towards carbapenems and other important AB classes, and their major AB resistance mechanism appears to be their acquisition and expression. Altogether, and underlying the One Health approach, the obtained results prove that the conventionally applied wastewater treatments are inefficient in the elimination of these microorganisms from the discharged effluents. These streams thus act as important vehicles for the dissemination of potentially pathogenic AB resistant bacteria and corresponding resistance genes into the environment, with a high potential to return to the human and animal populations by water or irrigated food consumption. Therefore, not only must the presence of AB resistant bacteria be monitored in the different environmental matrices, but targeted treatments should also be developed and implemented at full-scale in the WWTPs, so that the produced wastewater effluents could be safely discharged into the environment and/or reused for other purposes, such as agricultural irrigation. Additionally, and since this study was performed with samples collected from only one WWTP during a narrow period of time, studies with samples collected from multiple WWTPs in different time-points/seasons will be valuable to strengthen these findings and to answer to the new and exciting scientific question raised by this work: why is there a significant difference in the transmission of AB resistance genes between the groups of bacteria here distinguished as potentially environmental and potentially pathogenic?

## Figures and Tables

**Figure 1 antibiotics-10-01118-f001:**
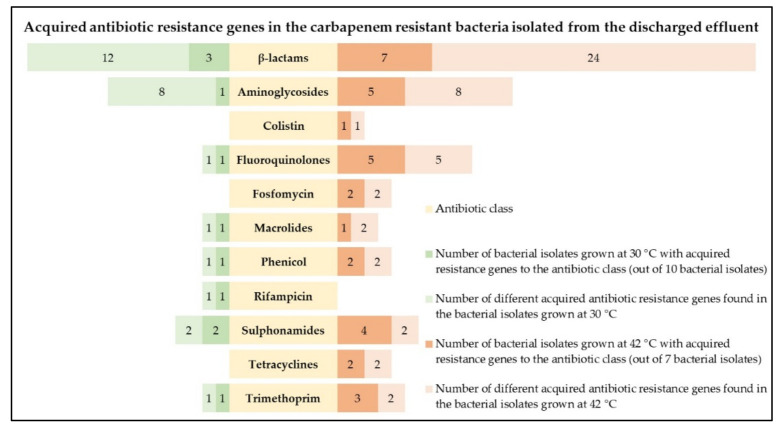
Number of bacterial isolates grown at 30 °C (green) and 42 °C (orange) from the discharged effluent samples harbouring acquired resistance genes towards different AB classes (yellow) and corresponding diversity of acquired AB resistance genes (light green and light orange).

**Figure 2 antibiotics-10-01118-f002:**
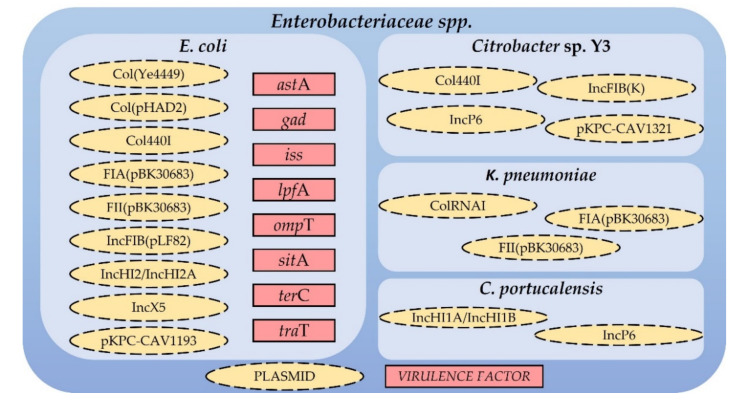
Plasmids found in the whole genomes of the carbapenem resistant *Enterobacteriaceae* isolated from the discharged effluent samples and virulence factors found in the *E. coli* isolate.

**Table 1 antibiotics-10-01118-t001:** AB resistance phenotype of the potentially pathogenic carbapenem resistant bacteria isolated from the discharged effluent samples.

Bacteria	AMP10	CTX5	IPM10	MEM10	CIP5	C30	CN10	TE30	W5	SXT25
*Acinetobacter baumannii*	R	R	R	R	R	R	R	R	R	S
*Acinetobacter pittii*	R	R	I	I	R	R	R	R	R	S
*Acinetobacter oleivorans*	R	R	I	I	R	R	S	R	R	R
*Aeromonas caviae*	*	*	*	*	S	*	*	*	*	R
*Aeromonas veronii* (1)	*	*	*	*	R	*	*	*	*	R
*Aeromonas veronii* (2)	*	*	*	*	R	*	*	*	*	S
*Citrobacter portucalensis*	R	R	R	R	R	R	R	R	R	R
*Citrobacter* sp. Y3	R	R	R	R	R	R	R	R	S	S
*Escherichia coli*	R	R	R	R	R	S	R	R	R	R
*Klebsiella pneumoniae*	R	R	R	R	R	S	R	R	R	R

AMP10—Ampicillin 10 µg; CTX5—Cefotaxime 5 µg; IMP10—Imipenem 10 µg; MEM10—Meropenem 10 µg; C30—Chloramphenicol 30 µg; CIP5—Ciprofloxacin 5 µg; CN10—Gentamicin 10 µg; TE30—Tetracycline 30 µg; W5—Trimethoprim 5 µg; SXT25—Trimethoprim + sulphamethoxazole 25 µg. * No EUCAST breakpoints available; S—Susceptible, standard dosing regimen; I—Susceptible, increased exposure; R—Resistant.

## Data Availability

The whole genome sequencing data files were deposited in GenBank within the BioProject with the SRA accession number PRJNA683808.

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
