# Peer review of "Environmental and Pathogenic Carbapenem Resistant Bacteria Isolated from a Wastewater Treatment Plant Harbour Distinct Antibiotic Resistance Mechanisms"

_antibiotics, 2021, doi:10.3390/antibiotics10091118_

Round 1
Reviewer 1 Report
In their manuscript "Environmental and pathogenic carbapenem resistant bacteria isolated from a wastewater treatment plant harbour distinct antibiotic resistance mechanisms" the authors describe a study conducted on samples of the water influx and efflux of a large-scale water treatment plant. According to the work of Hrenovic et al., the cultivation temperature of isolated bacteria allows the selective enrichment of carbapenem-resistant strains with presumable environmental origin (37 °C) and pathogenic ones (42 °C). Following that thesis, the authors demonstrate an increased prevalence of AB-resistance genes for bacteria cultivated at increased temperatures. Selected isolates were further subjected to whole-genome sequencing and AB-resistance profiling to characterize the discharged effluent more deeply.
The results of this study are of high relevance considering the increasing resistance against last-line antibiotics of the carbapenem or ß-lactam class and the data emphasizes the particular contribution of resistance acquired by horizontal gene transfer. However, to become a valuable contribution to the recent literature, some points should be improved:
Major comments:
- The authors indicate, that intrinsic mechanisms of resistance are the main drivers of resistance in the environmental strains but this is not shown directly. Either, the statement needs to be weakened, or this should be proven e.g. by a meta-transcriptomics / -proteomics study or quantification of the corresponding gene products from isolated cultures after AB treatment.
- According to the introduction, in the work developed by Hrenovic et al., the isolation was carried out at 37 and 42 °C, while in the present study 30 °C was chosen as lower temperature. The discrepancy should be explained.
- Please ensure data availability prior publication. Currently a BioProject with the accession number PRJNA683808 is not publicly available.
Minor comments:
- As the prevalence of the "blaXYZ" genes is worth an entire section in the results part, they should be described shortly in the introduction.
- The calculations used for figure 1 and supplemental table S1 are not entirely correct: The three plates, counted for each biological replicate, represent technical replicates and should exhibit a vastly different variance than the biological replicates. Thus the standard deviation must be calculated from the three average values of the biological replicates and not from all nine values to avoid mixing the levels of confidence. When doing so, the SD is slightly higher for all cases, e.g. the result for 30 °C effluent would be 5.37e3 ± 1.74e3 instead of 5.37e3 ± 1.58e3.
- The informative content of figure 1 could be increased by adding summarized information from section 2.2 (number of identified taxa and kind of/number of detected resistance genes) to it.
- The last sentence of section 2.3 (KmerFinder 3.2 tool being the most accurate tool for the identification of different bacterial strains) is missing a reference.
- To improve readability, section 2.4.1 should be streamlined a lot. The described information is contained in tables 1 & 2 and a statement that in the discharged effluent, incubated at 30 °C AB resistance genes were present in only three out of 10 bacteria, while at 42 °C such genes were detected in all seven sequenced strains, is getting lost in the current form.
- Taking into account typical article length limitations, the authors could also consider to provide tables 1 & 2 as supplemental information and replace both by one (schematic) representation in the main article.
- The description of table 3 should reference section 4.7 to explain the abbreviated AB names in the header.
- While the reported results are likely representative for the situation at various wastewater-treatment plants, they are obtained from a single sampling site during a narrow period of time. Therefore recalling this point and describing the actual sample origin would be valuable during the discussion.
- The pore-size stated in section 4.2 is likely 0.22 µm, not µM.
- The conclusion is very general and seems a bit unrelated to the present results. Moreover, the conclusion does not explain, why the authors conducted additional experiments after the initial screening for antibiotic resistance genes in the isolated cultures.
Author Response
Major comments:
- The authors indicate that intrinsic mechanisms of resistance are the main drivers of resistance in the environmental strains, but this is not shown directly. Either, the statement needs to be weakened, or this should be proven e.g., by a meta-transcriptomics / -proteomics study or quantification of the corresponding gene products from isolated cultures after AB treatment.
R: The statement as been changed accordantly.
- According to the introduction, in the work developed by Hrenovic et al., the isolation was carried out at 37 and 42 °C, while in the present study 30 °C was chosen as lower temperature. The discrepancy should be explained.
R: An explanation for this discrepancy was added in the Material and Methods section, in point 4.2.
- Please ensure data availability prior publication. Currently a BioProject with the accession number PRJNA683808 is not publicly available.
R: An email asking to release ASAP the data was sent to GenBank.
Minor comments:
- As the prevalence of the "blaXYZ" genes is worth an entire section in the results part, they should be described shortly in the introduction.
R: A sentence was added to the introduction section describing these genes.
- The calculations used for figure 1 and supplemental table S1 are not entirely correct: The three plates, counted for each biological replicate, represent technical replicates and should exhibit a vastly different variance than the biological replicates. Thus, the standard deviation must be calculated from the three average values of the biological replicates and not from all nine values to avoid mixing the levels of confidence. When doing so, the SD is slightly higher for all cases, e.g., the result for 30 °C effluent would be 5.37e3 ± 1.74e3 instead of 5.37e3 ± 1.58e3.
R: The calculations were recalculated accordantly and changed in supplementary table 1 and figure 1.
- The informative content of figure 1 could be increased by adding summarized information from section 2.2 (number of identified taxa and kind of/number of detected resistance genes) to it.
R: Figure 1 was altered to be more informative along with its legend.
- The last sentence of section 2.3 (KmerFinder 3.2 tool being the most accurate tool for the identification of different bacterial strains) is missing a reference.
R: We have introduced the missing reference.
- To improve readability, section 2.4.1 should be streamlined a lot. The described information is contained in tables 1 & 2 and a statement that in the discharged effluent, incubated at 30 °C AB resistance genes were present in only three out of 10 bacteria, while at 42 °C such genes were detected in all seven sequenced strains, is getting lost in the current form.
R: Section 2.4.1 was altered as suggested to highlight the AB resistance genes identified in the bacteria isolated at 30 °C and 42 °C.
- Taking into account typical article length limitations, the authors could also consider to provide tables 1 & 2 as supplemental information and replace both by one (schematic) representation in the main article.
R: Table 1 and 2 were removed as suggested and schematic figure was added.
- The description of table 3 should reference section 4.7 to explain the abbreviated AB names in the header.
R: This was changed as suggested.
- While the reported results are likely representative for the situation at various wastewater-treatment plants, they are obtained from a single sampling site during a narrow period of time. Therefore, recalling this point and describing the actual sample origin would be valuable during the discussion.
R: We have added that information to the discussion section.
- The pore-size stated in section 4.2 is likely 0.22 µm, not µM.
R: We have corrected that.
- The conclusion is very general and seems a bit unrelated to the present results. Moreover, the conclusion does not explain, why the authors conducted additional experiments after the initial screening for antibiotic resistance genes in the isolated cultures.
R: The conclusion was changed accordingly.
Reviewer 2 Report
Please see attached document

Author Response
Reviewer 2
Major comments:
1) Was the growth temperature the only criteria considered in grouping the isolated bacteria as potentially pathogenic or potentially environmental? I think that while this methodology is useful, it is not 100% specific, so possible biases should be included in the discussion.
R: An more detail explanation was added to the discussion section concerning the selection criteria.
2) I suggest the authors further discuss the genes detected in the isolates, since many of them represent a serious public health challenge, including the genes coding for carbapenems and colistin resistance.
R: We have discussed this issue in the discussion section as suggested.
3) The authors should explicitly state the selection criteria for the antimicrobials evaluated in the Kirby-Bauer tests.
R: A sentence explaining the selection criteria was added to the Material and Methods section.
4) I think that some of the key words are redundant, so I suggest to remove “pathogenic carbapenem resistant bacteria”, “intrinsic resistance”, and “acquired resistance”.
R: These key words were removed as suggested.
5) As the authors do not include any sample size calculation, they cannot determine prevalence. Therefore, they must refer to "detection rates" or “isolation rates”.
R: This was changed as suggested.
6) In my opinion, the description of the aim of the manuscript is not right, since some activities are considered as objectives. Additionally, this must be in past tense. So, I suggest the authors change it for: Therefore, the aim of this study was to characterize the phenotypic and genotypic antibiotic resistance of carbapenem resistant environmental and pathogenic bacteria isolated from a full-scale WWTP.
R: We have altered as suggested.
7) In my opinion, Figure 1 does not contribute to the expression of results, apart from that already included as text. Please remove it.
R: We have removed figure 1 and insert a new figure which is more informative.
8) Please adjust the format and size of Table 1.
R: We have changed it as suggested.
9) Lines 187-196: I think it would be much more explanatory to describe the identification rates of the different plasmids in general terms, and in Table 1 to indicate them in particular for each isolate. Otherwise, it is a bit confusing.
R: We have altered it as suggested.
Minor comments.
R: All the suggestions were taken into consideration and were removed, altered, or changed as suggested.
40) Lines 434-435: does the authors used any reference for this categorization or it was arbitrary?
R: We have added a reference.